# Learning Two-Step Hybrid Policy for Graph-Based Interpretable Reinforcement Learning

## Abstract

We present a two-step hybrid reinforcement learning (RL) policy that is designed to generate interpretable and robust hierarchical policies on the RL problem with graph-based input. Unlike prior deep reinforcement learning policies parameterized by an end-to-end black-box graph neural network, our approach disentangles the decision-making process into two steps. The first step is a simplified classification problem that maps the graph input to an action group where all actions share a similar semantic meaning. The second step implements a sophisticated rule-miner that conducts explicit one-hop reasoning over the graph and identifies decisive edges in the graph input without the necessity of heavy domain knowledge. This two-step hybrid policy presents human-friendly interpretations and achieves better performance in terms of generalization and robustness. Extensive experimental studies on four levels of complex text-based games have demonstrated the superiority of the proposed method compared to the state-of-the-art.

## 1 Introduction

Recent years have witnessed the rapid developments of deep reinforcement learning across various domains such as mastering board games (Silver et al., 2016; Schrittwieser et al., 2020) playing video games (Mnih et al., 2015), and chip design (Mirhoseini et al., 2021), etc. The larger and complicated architecture of deep reinforcement learning models empowered the capabilities of resolving challenging tasks while brought in significant challenges of interpreting the decision making process of those complex policies (Puiutta & Veith, 2020). This trade-off between performance and interpretability becomes an inevitable issue when DRL is applied to high stakes applications such as health care (Rudin, 2019). In this work, we focus on graph-based interpretable reinforcement learning (Zambaldi et al., 2018; Waradpande et al., 2020) as the graph representation is expressive in various domains including drug discovery (Patel et al., 2020), visual question answering (Hildebrandt et al., 2020), and embodied AI (Chaplot et al., 2020; Huang et al., 2019), etc. Another benefit of studying graph-based RL is that the graph structure can provide natural explanations of the decision-making process without the necessity of introducing new programms (Verma et al., 2018) or heavy domain knowledge (Bastani et al., 2018) for interpretation. Prior works in interpretable reinforcement learning (Verma et al., 2018; Madumal et al., 2020; Liu et al., 2018) either works on restricted policy class (Liu et al., 2018) that leads to downgraded performance, or the interpretablility (Shu et al., 2017; Zambaldi et al., 2018) is limited. Another common issue of interpretable RL is that the provided explanation is generally difficult to comprehend for non-experts (Du et al., 2019).

To resolve the challenges mentioned above, we propose a novel two-step hybrid decision-making process for general deep reinforcement learning methods with graph input. Our approach is inspired by the observation of human decision-making. When confront complicated tasks that involve expertise from multiple domains, we typically identify which domain of expert we would like to consult first and then search for specific knowledge or solutions to solve the problem. Recognizing the scope of the problem significantly reduces the search space of downstream tasks, which leads to a more simplified problem compared to find the exact solution in all domains directly. As an analogy of this procedure, we disentangle a complicated deep reinforcement learning policy into a classification the problem for problem type selection and rule-miner. The classification establishes a mapping from complex graph input into an action type, which handles high-order logical interactions among node and edge representations with graph neural network. The rule miner conducts explicit one-hop reasoning over the graph and provides user-friendly selective explanations (Du et al., 2019) by mining several

decisive edges. This two-step decision making is essential not only for providing interpretability, but also for generalization and robustness. It is intuitive to see that the simplified classification is easier to achieve better generalization and robustness than the original complicated RL policy. Furthermore, the rule-miner identifying key edges in the graph is much more robust to the noisy perturbations on the irrelevant graph components.

In summary, our contributions are three folds:

- We formalize an interpretable deep reinforcement learning problem based on graph input.
- We propose a two-step decision-making framework that achieves far better performance in terms of generalization and robustness and provides human-friendly interpretations.
- Extensive experiments on several text-based games (Côté et al., 2018) demonstrated that the proposed approach achieves a new state-of-the-art performance for both generalization and robustness.

## 2 RELATED WORKS

### 2.1 INTERPRETABLE RL

Instead of using deep RL as a black box, researchers also worked on making deep RL more interpertable. (Mott et al., 2019; Zambaldi et al., 2018) introduce some sorts of attention mechanisms into policy networks and explain decision making process by analyzing attention weights. (Verma et al., 2018) combines program synthesis and PIDs in classic control to solve continuous control problems, while keep the decision making interpretable by explicitly writing the program. Combining symbolic planning and deep RL has similar effects, like (Lyu et al., 2019). Tree-based policies (Bastani et al., 2018) are favorable in interpretable RL, since they are more human-readable and easy to verify. However, many existing works of interpretable RL sacrifices the performance for the interpretability. In contrast, our method achieves even better generalization and robustness while providing interpretability.

### 2.2 HIERARCHICAL RL

Hierarchical RL focuses on decomposing a long-horizon tasks into several sub-tasks, and applying a policy with hierarchical structure to solve the task. (Vezhnevets et al., 2017; Frans et al., 2018) try to solve generic long-horizon tasks with a two-level policy, where a high-level policy will generate a latent sub-goal for low-level policies, or directly select a low-level policy, and the low-level policy is responsible for generating the final action. In some robot locomotion tasks, people usually manually define the choices of sub-goals to boost the performance and lower the learning difficulty, like (Nachum et al., 2018; Levy et al., 2018). Our two-step hybrid policy is reminiscent of the two-level policy architecture in hierarchical RL. However, the high-level policy in hiereachical RL is mainly used to generate a sub-goal while the action pruner in our hybrid policy is used to reduce the number of action candidates. And the motivations of our method and hierarchical RL are also different: hierarchical RL targets at decomposing a long-horizon tasks into several pieces, but our method focuses on interpertability, generalizability and robustness.

### 2.3 REFACTORING POLICY

Given a trained neural network-based policies, one may want to refactor the policy into another architecture, so as to improve the generalizibility or interpretablity. And this refactoring processs is usually done by imitation learning. The desired architectures of the new policy can be decision tree (Bastani et al., 2018), symbolic policy (Landajuela et al., 2021), a mixture of program and neural networks (Sun et al., 2018), or graph neural networks (Mu et al., 2020). In our method, we also use similar techniques to refactor a reinforcement learning policy into a two-step hybrid policy.

### 2.4 COMBINE NEURAL NETWORKS WITH RULE-BASED MODELS

Neural networks and rule-based models excel at different aspects, and many works have explored how to bring them together, like (Chiu et al., 1997; Goodman et al., 1990; Ray & Chakrabarti, 2020;

Okajima & Sadamasa, 2019; Greenspan et al., 1992). Therefore various ways to combine neural networks with rule-based models. For example, (Wang, 2019) combines them in a "horizontal" way, i.e., some of the data are assigned to rule-based models while the others are handled by neural networks. In contrast, our work combines neural networks and rule-based models in a "vertical" way, i.e., the two models collaborate to make decisions in a two-step manner.

## 3 A GENERAL FRAMEWORK FOR TWO-STEP HYBRID DECISION MAKING

In this section, we first describe our problem setting including key assumptions and a general framework that formulates the decision-making process in a two-step manner.

We consider a discrete time Markov Decision Process (MDP) with a discrete state space $\mathcal{S}$ and the action space $\mathcal{A}$ is finite. The environment dynamics is denoted as $\mathbf{P} = \{p(s'|s, a), \forall s, s' \in \mathcal{S}, a \in \mathcal{A}\}$. Given an action set $\mathcal{A}$ and the current state $s_t \in \mathcal{S}$, our goal is to learn a policy $(\pi)$ that select an action $a_t \in \mathcal{A}$ to maximize the long-term reward $\mathbf{E}_\pi[\sum_{i=1}^{T} r(s_i, a_i)]$. Assume we are able to group the actions into several mutual exclusive action types $(A_k)$ according to its semantic meanings. More concretely, the $k$-th action type $A_k = \{a_k^1, ..., a_k^n\}$ denotes a subset of actions in original action space $A_k \subseteq A$. Then we have $A_1, A_2, .., A_K \subseteq \mathcal{A}, A_i \cap A_j = \emptyset (i \neq j), \cup_{i=1}^{K} A_i = \mathcal{A}$. It is worth noting that the number of action type $K$ is usually more than an order of magnitude smaller than original actions $(K \ll |\mathcal{A}|)$.

Let the policy $\pi = \langle f_p, f_s \rangle$ represented by a hybrid model that consists of action pruner $f_p$ and action selector $f_s$. The action pruner is used to prune all available action candidates to a single action type, i.e., $f_p(s_i) = k$, where $k$ is the index of chosen action type and $A_k \in \{A_1, A_2, .., A_K\}$. Then the action selector is used to select a specific action given the action type chosen by the action pruner, i.e., $f_s(s_i, A_k) = a_i$, where $a_i \in A_k$ and $k = f_p(s_i)$.

Intuitively, this design intends to disentangle different phases in decision-making process to two different modules. On one hand, determining the action type typically involves high-order relationships and needs a model with strong expressive power, then neural network is a good candidate in this regard. On the other hand, selecting an action within a specific action type can resort to rule-based approaches, which is essential in providing strong intrepretability, generalizability and robustness.

Figure 1 shows the overall pipeline of our two-step hybrid decision making. The agent will receive a state $s_i$ at each time step $i$. At time step $i$, we will first call the action pruner $f_p(s_i)$ to select the action type $A_i$. Then the rule-based action selector $f_s(s_i, A_i)$ will take as inputs the current state $s_i$ and the action type $A_i$ given by action pruner to select the specific action to be executed in this step.

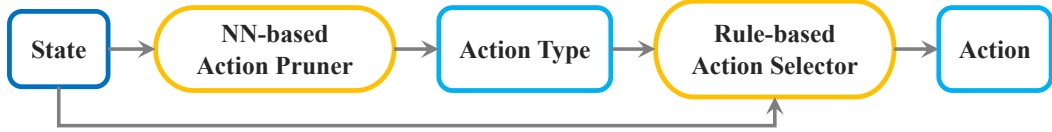

Figure 1: Overall pipeline of the two-step hybrid decision making.

## 4 TWO-STEP HYBRID POLICY FOR TEXT-BASED GAMES WITH GRAPH INPUT

### 4.1 OVERVIEW

In this section, we instantiate our proposed framework in the setting of text-based games (Côté et al., 2018) with graph input. In text-based games, the agent receives a knowledge graph (as shown in Figure 3) that describes the current state of the environment and the task to be completed, e.g., the task can be represented by several edges like ("potato", "needs", "diced"). Our goal is to learn a policy that maps the input knowledge graph to an action from the provided action set $\mathcal{A}$. Each action $a_j \in \mathcal{A}$ is a short sentence, e.g., "take apple from fridge".

In this setting, we use graph neural networks (GNNs) as the action pruner $f_p$, and a rule-based model as the action selector $f_s$. We will elaborate the details of the action pruner and the action selector in Sec 4.3 and Sec 4.4, respectively.

Training the GNN-based action pruner and the rule-based action selector by reinforcement learning is nontrivial since the whole pipeline is not end-to-end differential. Therefore, we propose to learn both models separately from a demonstration dataset, and the demonstration dataset can be obtained by a trained reinforcement learning agent. We will elaborate the details in Sec 4.2.1. This process is inspired by existing works like (Bastani et al., 2018; Landajuela et al., 2021; Sun et al., 2018; Mu et al., 2020), where policies trained by reinforcement learning can be refactorized into other forms.

## 4.2 Preparing Datasets for Learning Two-Step Hybrid Policy

### 4.2.1 Demonstration Acquisition

Given a set of interactive training environments, the goal of this step is to obtain a demonstration dataset, which contains state-action pairs from a policy achieving high reward in the training environment. Specifically, we target at generating a demonstration dataset $\mathcal{D} = \{(s_i, \pi(s_i))\}_{i=1}^{N}$, where $s_i$ is the input state (a knowledge graph in our cases) from the training environments, and $\pi(s_i)$ is the output of demonstration policy on the state $s_i$. The representation of $\pi(s_i)$ is flexible: it can be an action, the logits, or any latent representation, which indicates the demonstration action distribution. This dataset will be used for learning our two-step hybrid policy.

We refer the policy used to generate demonstration as *teacher policy*. It can be obtained in any way, as long as it provides reasonable and good supervision on how to solve the task in the training environments, and it is not necessary to have strong interpretablity, generaliziabilty or robustness. Therefore, deep reinforcement learning is a great tool to learn such a policy.

After obtained the teacher policy, we use it to interact with the training environments to collect states and label them with the output of the teacher policy. During the interaction, we add action noise to perturb the state distribution in demonstration dataset. A more diversified state distribution is beneficial to the performance of our two-step hybrid approach.

### 4.2.2 Action Grouping and Demonstration Splits

As mentioned above, the action pruner is responsible for selecting the correct type of actions for the current state, so that the action candidates will be pruned to a smaller set. Then, a rule-based model will work on this specific action type to select the action to be executed. Therefore, we needs to label each state with the action type. Since the action for each state is already in the dataset, we just need to convert each action into an action type. In other words, we need to group all possible actions into several action types.

In text-based games, each action is a short sentence with semantic meaning, e.g., "take apple from fridge". Intuitively, we can group the actions by their semantic meaning. For example, if the actions are "take apple from fridge", "slice potato with knife", "dice cheese with knife", "cook carrot with oven", and "cook onion with stove", we can group them into three types: "take" actions, "cut" actions, and "cook" actions. And we make sure all the actions with the same action type follow the same template, e.g., the template of "take" action can be written as "take object from receptacle", where object and receptacle can be instantiated by appropriate words.

Grouping actions by semantic meanings also aligns with our motivation: 1) After we obtained $K$ group of actions, we substantially alleviate the workload of neural networks from learning a policy that maps states to an action in a varying size and high-dimensional action space to a simplified classification with a much smaller number of classification categories. This simplification could greatly reduces the sample-complexity. 2) Furthermore, this step also eliminates the search space to enable a feasible candidates set for the rule-miner.

Formally, given a dataset $\mathcal{D} = \{(s_i, \pi(s_i))\}_{i=1}^{N}$, we will generate an action type for each state, policy output $(s_i, \pi(s_i))$ pair. The action type $k_i$ is determined by $k_i = h(\pi(s_i))$, where $h(\pi(s_i))$ is a domain-specific function which takes the semantic meaning of the action into consideration. As a result of the clustering process, we will get a new demonstration dataset with action types $\mathcal{D} = \{(s_i, \pi(s_i), k_i)\}_{i=1}^{N}$. In addition, we can also split the demonstration dataset based on the action types to get $K$ subset of the original demonstration dataset, $\{\mathcal{D}_1, \mathcal{D}_2, ..., \mathcal{D}_K\}$, where $\mathcal{D}_k = \{s_i, \pi(s_i)|h(\pi(s_i)) = k\}$.

### 4.3    GNN-BASED ACTION PRUNER

The action pruner needs to output an action type based on the input knowledge graph, so it is essentially a classifier. Given the demonstration dataset $\mathcal{D} = \{(s_i, k_i)\}_{i=1}^N$ obtained in the last step, we want to train a classifier $f_p(s; \theta) = k$, where $k \in \{1, 2, ..., K\}$ is an action type. This is a conventional classification problem which can be solved by minimizing cross entropy loss:

$$\theta = \arg\min_\theta - \sum_i \sum_{j=1}^K k_i^j \log(f_\theta^j(s_i)),$$

where $f_\theta(s_i)$ outputs a probability distribution over the $K$ action types, $f_\theta^j(s_i)$ denotes the $j$-th action type's probability. $k_i^j \in \{0, 1\}$ denotes denotes whether the action type $j$ was chosen in the demonstration dataset at state $i$.

### 4.4    RULE-BASED ACTION SELECTOR

#### 4.4.1    ABSTRACT SUPPORTING EDGE SETS

When the input is a knowledge graph, the action is naturally strongly correlated with some critical edges. For example, ("potato", "needs", "diced") and ("potato", "in", "player") can lead to the action "dice potato". We refer those decisive edges correlating to an action as the *supporting edge set* of this action. Since we have grouped actions by their semantic meanings, actions within each action type are actually supported by similar edges. For example, "dice potato" is supported by ("potato", "in", "player"), and "dice apple" is supported by ("apple", "in", "player"). As mentioned in Sec 4.2.2, each action type comes with an action template like "dice object". Based on the action template, we can perform some sorts of abstraction. For example, given an input knowledge graph labeled with action "dice apple", we can replace all the "apple" appearing in the graph edges and the action with an abstract name "object". Then we can say the action "dice object" is essentially supported by ("object", "in", "player"), where the two "object" should be instantiated by the same word. Under this kind of abstraction, different actions within the same action type can share a same *abstract supporting edge set* which contains edges with abstract names.

The *abstract supporting edge set* indicates the decisive edges for an action type, and it can be instantiated for each specific action. For example, to check whether the action "dice apple" should be executed, the abstract edge ("object", "in", "player") will be instantiated to edge ("apple", "in", "player"). Then, the existence of ("apple", "in", "player") in input knowledge graph becomes an evidence for selecting the action "dice apple". We aim at finding an abstract supporting edge set for each action type, and it will be used during inference.

#### 4.4.2    MINE ABSTRACT SUPPORTING EDGE SETS FROM DEMONSTRATIONS

Finding the abstract supporting edge set for each action type is actually a rule mining process. There are several off-the-shelf rule miners like FP-Growth (Han et al., 2004), Apriori (Agrawal et al., 1994), Eclat (Zaki, 2000), etc. , but they are not designed for knowledge graphs. Thus, we propose a simple yet effective rule miner for our setting to discover the supporting edge sets.

To find the **A**bstract **S**upporting **E**dge set ASE$(A_k)$ for each action type $A_k$, we designed a numerical statistic that is intended to reflect the importance of an edge when taking an action $a \in A$, and this numerical statistic is inspired by tf-idf (Rajaraman & Ullman, 2011). Formally, for action type $A_k$, we have a subset of demonstration dataset $\mathcal{D}_k = \{s_i\}$ (ignoring $\pi_i$ here). Under the abstraction mentioned above, we can count the edge frequency for every (abstract) edge $e$ in $\mathcal{D}_k$:

$$freq_k(e) = \frac{|\{s_i | s_i \in \mathcal{D}_k, e \in s_i\}|}{|\mathcal{D}_k|}.$$

Similarly, we can also count the edge frequency for the entire demonstration dataset:

$$freq(e) = \frac{|\{s_i | s_i \in \mathcal{D}, e \in s_i\}|}{|\mathcal{D}|}.$$

And we can define an importance score of an edge w.r.t to the action type $A_k$:

$$I_k(e) = freq_k(e) \cdot \log(\frac{1}{freq(e)}),$$

where the term $freq_k(e)$ is similar to the term-frequency (tf), and the term $log(\frac{1}{freq(e)})$ is similar to the inverse document frequency (idf).

Then we can get the $\text{ASE}(A_k)$ by selecting the edges with the importance higher than a threshold, i.e., $\text{ASE}(A_k) = \{e | I_a(e) > \tau\}$, where $\tau$ is a hyperparameter shared across all action types.

### 4.4.3 INFERENCE BASED ON SUPPORTING EDGES

During inference, we use the supporting edge sets to score each action within the action type provided by action pruner, and select the action with the highest score. Figure 2 shows a concrete example of using supporting edge set to score an action.

Firstly, the action pruner outputs an action type based on the input knowledge graph, and we can retrieve the abstract supporting edge set of this action type. Secondly, given an action within the action type, we can instantiate the abstract supporting edge set to a specific supporting edge set by replacing the abstract names by the concrete words according to the action, e.g, ("object", "in", "player") will be instantiated to ("potato", "in", "player") if the action is "cook potato with oven". Finally, we can compare the input knowledge graph with the supporting edge set of each action to figure out which supporting edge set is best covered by the input knowledge graph. The number of overlapped edges between the supporting edge set and the the input knowledge graph will be regarded as the score of the action.

Formally, the inference process of our rule-based action selector can be described as follows

$$f_s(s, A) = \arg\max_{a \in A} |s \cap \text{SE}(a)|,$$

where $\text{SE}(a)$ is the supporting edge set associated with action $a$, which is obtained by instantiating the abstract supporting edge set of action type $A$.

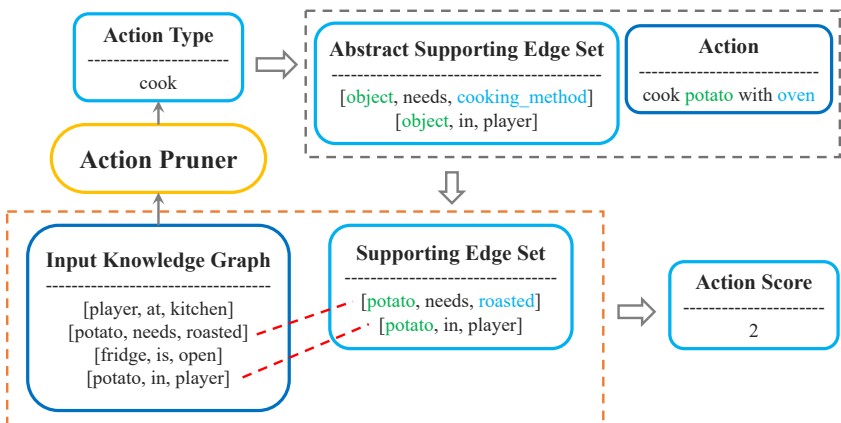

Figure 2: An example of using supporting edge set to score an action. Based on the knowledge graph, the action pruner first predict the action type (e.g., cook) that needs to be taken at current state, then instantiate the abstract supporting edge set to a concrete supporting edge set. Comparing the input knowledge graph with the supporting edge set, we can compute action score for each action and select the action with highest score accordingly.

## 5 EXPERIMENTS

### 5.1 DATASET SETUP

We evaluate our method on TextWorld, which is a framework for designing text-based interactive games. More specifically, we use the TextWorld games generated by GATA (Adhikari et al., 2020). In these games, the agent is asked to cook a meal according to given recipes. It requires the agent to navigate among different rooms to locate and collect food ingredients specified in the recipe, process the food ingredients appropriately, and finally cook and eat a meal.

| Difficulty Level | Recipe Size | Number of Locations | Need Cut | Need Cook | Number of Action Candidates | Number of Objects |
|---|---|---|---|---|---|---|
| 1 | 1 | 1 | Yes | No | 11.5 | 17.1 |
| 2 | 1 | 1 | Yes | Yes | 11.8 | 17.5 |
| 3 | 1 | 9 | No | No | 7.2 | 34.1 |
| 4 | 3 | 6 | Yes | Yes | 28.4 | 33.4 |

Table 1: TextWorld games statistics (averaged across all games within a difficulty level). The games and statistics are generated by (Adhikari et al., 2020).

The state received by the agent is a knowledge graph describing all the necessary information about the game. All the nodes and edges in the knowledge graph are represented in text. Figure 3 shows a partial example of input knowledge graph. The actions are also represented in text. Note that the number of available actions vary from state to state, so most of of existing network architecture used deep reinforcement learning cannot be directly used here.

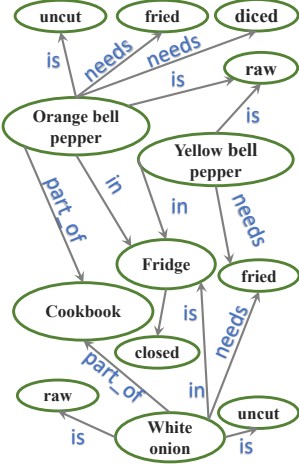

The games have four different difficulty levels, and each difficulty level contains 20 training, 20 validation, and 20 test environments, which are sampled from a distribution based on the difficulty level. The higher the difficulty levels are, the more complicated recipe will be and the more rooms food ingredients will be distributed among. Statistics of the games are shown in Table 1. For evaluating model generalizability, we select the top-performing agent on validation sets and report its test scores; all validation and test games are unseen in the training set. [1]

Figure 3: Visualization of a part of an input knowledge graph in TextWorld.

## 5.2 Implementation Details

### 5.2.1 Process Knowledge Graphs by GNNs

Graph neural networks (GNNs) serve as backbone networks in both our RL teacher policy and the action pruner in our two-step hybrid policy. More specifically, we use Relational-GCN to take edge attributes into consideration.

The relations and nodes of the input knowledge graphs are initially represented in text (words or short sentences). We use fastText (Mikolov et al., 2017), pre-trained on Common Crawl, to convert each token to an embedding, and then average all the token embeddings to obtain the mean fastText embedding of the entire text string. These mean fastText embeddings are pre-computed and fixed during training. For each relation and node, we append their corresponding mean fastText embedding with a trainable embedding vector. The concatenated embeddings are then used as the numerical representation of each relation and node, which are further passed through GNNs to get the latent embedding of the entire input knowledge graph.

### 5.2.2 Demonstration Acquisition

To collect demonstration dataset, we first train a teacher policy by DQN (Mnih et al., 2015) in the training environments, which can converge to a near-optimal solution. To adopt variable number of available actions, we let the policy network take the state and an available action as input and output a score for this action. And we can select the action with maximum score as the final output.

The trained teacher policy is used to collect 300K samples through the interaction with the environment, and label them with the taken actions, as illustrated in Sec 4.2.1. When collecting the demonstration dataset, we use $\epsilon$-greedy exploration strategy to increase the diversity of states.

---

[1]The code to reproduce all results in this work will be publicly available.

| | Action Type: Cut | | | | Example Action | | |
|---|---|---|---|---|---|---|---|
| Action | verb object with knife | | | | slice potato with knife | | |
| | Head | Relation | Tail | Importance | Head | Relation | Tail |
| Discovered Supporting Edges | object | in | player | 1.7004 | potato | in | player |
| | object | needs | verb_passive | 1.6659 | potato | needs | sliced |
| | object | part_of | cookbook | 1.0058 | potato | part_of | cookbook |
| | object | is | uncut | 0.9837 | potato | is | uncut |

Table 2: Discovered abstract supporting edges of the "cut" action type in difficulty level 1 of TextWorld environments. The importance of each edge is computed by the metric mentioned in Sec 4.4.2. The right column gives an example action and the corresponding supporting edge set, which is obtained by instantiating the abstract supporting edge set. The underlined words are abstract names for some certain words. For example, "verb" in action can be instantiated to "slice", "dice" or "chop", and "verb_passive" can be instantiated to "sliced", "diced" or "chopped" accordingly.

## 5.3 RESULTS

### 5.3.1 INTERPRETABILITY

The interpretablity of our two-step policy is two-fold: 1) the transparent two-step decision making process; 2) the rule-based models make decisions in a way which is easy to interpret by human.

Table 2 shows some representative rules discovered by our rule miners. We observed that all of the four discovered abstract supporting edges are indeed prerequisites of the "cut" actions. In particular, the abstract supporting edge ("object", "needs", "verb_passive") is a crucial prerequisite for the agent to select the correct verb in the "cut" actions. For example, if the input knowledge graph contains the edge ("potato", "needs", "sliced"), then the action "slice potato with knife" will get one more score than others because this edge is in the supporting edge set of action "slice potato with knife".

Since we clearly see the rule-based model makes decisions based on these rules, it is not hard to check whether this model works in a correct way. In this way, the agent can certainly select the correct "cut" action even in unseen test environments.

### 5.3.2 GENERALIZATION

| | Training | | | | Test | | | |
|---|---|---|---|---|---|---|---|---|
| Difficulty | 1 | 2 | 3 | 4 | 1 | 2 | 3 | 4 |
| GATA-GTF | 98.6 | 58.4 | 95.6 | 36.1 | 83.8 | 53 | 33.3 | 23.6 |
| Vanilla RL | 100 | 100 | 98.3 | 100 | 83.8 | 68 | 50 | 30.9 |
| Ours | 100 | 100 | 100 | 65.5 | **100** | **100** | **51.7** | **49.7** |

Table 3: Evaluation results on both training environments and test environment in TextWorld. The numbers show the agent's normalized scores.

We compare with two baselines: GATA-GTF and vanilla RL. GATA-GTF is a variant from GATA (Adhikari et al., 2020), and it uses the same ground-truth graph input as us. GATA-GTF processes the input graphs by relational graph convolutional networks (R-GCNs) (Schlichtkrull et al., 2018), and the whole pipeline is trained by DQN (Mnih et al., 2015). Vanilla RL is essentially the same method with GATA-GTF, but implemented by ourselves. With better implementation and careful hyperparameter tuning, it performs better than the implementation released by the GATA authors. And vanilla RL also serves the teacher policy which used to generate demonstrations for our method.

Table 3 shows the normalized scores of different methods on both training environments and test environment in TextWorld. The result of GATA-GTF is obtained from its paper (Adhikari et al., 2020). In all the environments, our agent achieves better generalization performance the vanilla RL (which is our RL teacher) and GATA-GTF baselines. Our agent can generalize pretty well to the

| Noise | | Difficulty | | | | | | | |
|---|---|---|---|---|---|---|---|---|---|
| | | 1 | | 2 | | 3 | | 4 | |
| Add | Drop | RL | Ours | RL | Ours | RL | Ours | RL | Ours |
| 0.2 | 0 | **100(0%)** | **100(0%)** | **100(0%)** | **100(0%)** | 96(-1%) | **98(-1%)** | 38(-61%) | **64(-1%)** |
| 0.2 | 0.03 | **100(0%)** | 96(-3%) | 81(-19%) | **98(-1%)** | 80(-18%) | **98(-1%)** | 41(-58%) | **51(-21%)** |
| 0.2 | 0.06 | **100(0%)** | 92(-7%) | **93(-6%)** | 82(-18%) | 86(-11%) | **93(-6%)** | **44(-55%)** | 41(-36%) |
| 0.4 | 0 | **100(0%)** | **100(0%)** | 85(-15%) | **100(0%)** | 65(-33%) | **91(-8%)** | 19(-80%) | **58(-11%)** |
| 0.4 | 0.03 | **100(0%)** | **100(0%)** | 77(-23%) | **90(-9%)** | 71(-27%) | **91(-8%)** | 20(-79%) | **53(-18%)** |
| 0.4 | 0.06 | **100(0%)** | 96(-3%) | 78(-22%) | **83(-16%)** | 65(-33%) | **96(-3%)** | 19(-80%) | **49(-24%)** |
| 0.6 | 0 | **100(0%)** | **100(0%)** | 85(-15%) | **100(0%)** | 76(-22%) | **93(-6%)** | 8(-91%) | **59(-9%)** |
| 0.6 | 0.03 | **100(0%)** | 97(-2%) | 75(-25%) | **83(-16%)** | 61(-37%) | **86(-13%)** | 9(-90%) | **39(-39%)** |
| 0.6 | 0.06 | **100(0%)** | 91(-8%) | 76(-24%) | **80(-20%)** | 60(-38%) | **96(-3%)** | 10(-89%) | **46(-29%)** |

Table 4: Robustness analysis of our method and vanilla RL baseline. We evaluate the performance of agents on **training environments** under different noise levels, e.g., "add 0.2 drop 0.03" means we randomly add 20% additional edges while randomly dropping 3% existing edges in graph. The numbers out of parentheses are normalized scores and the numbers in parentheses are relative performance change comparing to the performance without input noise. The bold numbers indicate which method performs better in each setting.

unseen test environments in all difficulty levels, while the performance of GATA-GTF and vanilla RL performs poorly in unseen test environments.

### 5.3.3 ROBUSTNESS

As mentioned above, our method also aims at robustness. To evaluate the robustness of our models, we add different levels of noises to input knowledge graphs and regard the performance of the agents under noisy inputs. In this paper, we define the noise on a knowledge graph as adding additional edges to the graph or dropping existing edges in the graph. Formally, we add noise at $(k, p)$-level to input knowledge graph in the following way:

1. Add edges: $\lceil k * |E| \rceil$ additional edges will be randomly generated and added to the graph, where $E$ is the edge set of the graph. For each edge $h_i, t_i, r_i$, head node $h_i$ and tail node $t_i$ are sampled from $V_{all}$ and relation $r_i$ is sampled from $R_{all}$, where $V_{all}$ means the set of all possible nodes and $R_{all}$ is the set of all possible relations.

2. Drop edges: $\lceil p * |E| \rceil$ edges will be dropped from the graph. Note that only original edges will be dropped, and the randomly added edges will not be dropped.

Table 4 shows the performance of vanilla RL and our method under noisy input graphs generated in the above mentioned way. Note that we test robustness in training environments instead of test environments, because the performance of both agents in test environments are not good enough and examining robustness of poor-performed agents does not make too much sense.

We observed that in difficulty level 1, both agents are very robust under the noisy inputs. However, in difficulty level 2 & 3 & 4, the performance of the RL agents are hurts a lot by the input noises, especially in the difficulty level 4. In contrast, the agents obtained by our method are still performs pretty good and the performance drops significantly less than the RL agents.

## 6 CONCLUSION

In this work, we propose a two-step hybrid policy for graph-based reinforcement learning. The two-step hybrid policy disentangles complicated black-box deep reinforcement learning policy into an action pruner and an action selector. The joint effort of two modules can not only generate human-friendly explanations of each decision, but also provide significantly better performance in terms of generalization and robustness. We conduct comprehensive experiments on the representative text-based environments (Côté et al., 2018) to demonstrate the effectiveness of our approach.

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
