# OpenReview forum: "Learning Two-Step Hybrid Policy for Graph-Based Interpretable Reinforcement Learning"
_ICLR.cc/2022/Conference — ICLR 2022 Submitted_

### Official Review · Reviewer_3CUH · 2021-11-02

**Correctness:** 4
**Technical Novelty And Significance:** 2
**Empirical Novelty And Significance:** 2
**Recommendation:** 3
**Confidence:** 4

**Main Review:**

In general, I am worried that this framework is not very generalizable outside of this specific problem.

First, as the authors acknowledge, it only makes sense for graph-based inputs where there is no real perception. The input just tells you what objects are available. This method would not work at all in any other kind of environment, such as one with visual perception instead of symbolic input, in continous/physics based environments, or even games such as atari where there aren't symbolic inputs and graphs. This is already a pretty narrow problem.

On top of that, this method is only tested on one extremely specific subset of all graph symbolic-input environments relating to cooking on a single dataset. And cooking seems to be an extremely limited domain (mostly a fixed set of the kinds of actions you can do with a relatively small set of objects in a small space). On average, how many steps are required for each of these difficulties? With the graph input nature of the dataset and how many steps are required in the recipe, it seems like the GT policy would be really really short, right? Does this method work on other TextWorld games, perhaps ones with more different kinds of actions? Or environments that also have graph inputs such as VirtualHome?

There is also a detail that is really briefly mentioned in the paper that is concerning. Specifically what is the method is selecting this set of K different action types and how the ground truth label k_i is selected for each base action? The paper says that "the action type k_i is determined by k_I = h(\pi(s_i)) where h is a domain specific function which takes the semantic meaning of the action into consideration." This is not at all clear, but it sounds like it's saying that this is a hand-crafted rule by the designed. In which case, I would argue that is putting way too much of the specifics of the environment as hand-crafted features. This means that if we have another environment, for instance a graph-based battle simulator in textworld that a human designer would need to go in and hand-craft parsing rules for all of the possible actions the agent could do in that environment and so on and so on for any new task you want to add. This seems very unscalable and putting a heavy thumb on the scale.

This action grouping idea also seems quite specific to text adventure games or environments where the action spaces is language. This grouping actions by their specific type does not seem to really occur in any other environment I can think of other than text-based or purely symbolic environments.

There are by this point very good methods that show generalization results on all of TextWorld and on other text adventure games. With train/test splits occurring across completely different categories of games, and without the restriction of only graph inputs. There are much more general methods which perform extremely well in these environments, so I am not entirely sure how useful a line of work is on this very narrow subproblem.

For evaluation, there should also be more comparison to state-of-the-art methods in TextWorld. GATA-GTF is a good baselines because it also was developed on this specific subset of TextWorld, and the RL baseline is also good. But there are many other methods on TextWorld that also do not have this constraint on graph inputs and are demonstrated on all of TextWorld. There should be at least one or two comparisons to one of these SOA methods.

Minor: Is the Vanilla RL actually the same as the policy used by the method to obtain the teacher policy? If not, could you show the results of the teacher policy in Table 3, that seems like an interesting comparison to make and it's already available.

On the positive side, I did find the edge set mining part of the method interesting, although I'm not sure whether it is a method that is very applicable in non-graph-based inputs.


**Summary Of The Paper:**

This paper introduces a two-step method for RL problems with graph-based inputs. This method trains a teacher policy which it then uses to construct a policy "pruner" which predicts the kind of action needed. It then uses a ideas from retrieval to select the edges needed to complete the action.

**Summary Of The Review:**

The method is interesting, but it seems very contrived for a very small and specific type of problem. There appear to be a lot of components which would not generalize even to very similar TextWorld domains. The result is only tested on one small subset of TextWorld and the state-of-the-art comparisons are not adequate.

---

### Official Review · Reviewer_eXdP · 2021-11-02

**Correctness:** 3
**Technical Novelty And Significance:** 2
**Empirical Novelty And Significance:** 3
**Recommendation:** 5
**Confidence:** 3

**Details Of Ethics Concerns:**

not applicable.

**Main Review:**

Positive points of the paper: (i) use of more powerful representations (graphs) to describe the task and states; (ii) generating decisions that can be explained in a human-friendly representation; (iii) dividing the problem into two levels (first mapping the states into abstract actions and then mapping these actions into primitive actions); (iv) interesting proposal to mine the ASEs (GNN-Abstract Supporting Edge sets) from the demonstrations given by the teacher.
(v) good results in complex text-based games; (vi) clear and well-written paper.

Negative points of the paper: (i) why does the paper refer to interpretable RL (even in the title), since the proposal is basically a combination of supervised (SL) and unsupervised (UL) learning? (ii) The related works do not, in fact, provide an overview of decision-making techniques that combine SL and UL. (iii) It wasn't clear to me how h(.) is defined (function that groups actions) - the definition of this function seems to be quite manual and domain dependent, which restricts its use of the proposal. (iv) It wasn't clear to me exactly what the inputs are: In the pipeline in Fig1 it indicates that the state (given in knowledge graph - KG) is the input, but in the text, it indicates that KG also describes the task. Clarify this further in the text.


**Summary Of The Paper:**

This paper proposes a pipeline to determine (primitive) actions to be performed in states described by knowledge graphs. The pipeline is divided into two steps. The first step, Action Pruner, maps states into abstract actions (i.e., clusters of primitive actions) and is performed by a trained GNN with data from a teacher. The second step, Action Selector, is a rule-based system that defines the primitive action to be performed among those included in the abstract action given by the previous step. The Action Selector defines the primitive action based on the GNN Abstract Supporting Edge set (ASE), also defined from the training data provided by the teacher. The proposal presented good results in games with text-based environments and also allows the generated decisions to be human-friendly explanations.

**Summary Of The Review:**

The paper reports a method that decides an action (textual) based on a state (and task) described by a knowledge graph. The idea is interesting, and the results are significant, in addition to enabling the interpretation of decisions made by the system. However, the scope of the proposal seems limited and, despite being well written, some points of the proposal should be further clarified.

---

### Official Review · Reviewer_5Hfh · 2021-11-02

**Correctness:** 4
**Technical Novelty And Significance:** 2
**Empirical Novelty And Significance:** 3
**Recommendation:** 5
**Confidence:** 4

**Main Review:**

## What I like about this paper

- I like to see research on RL applied to graph data. I believe several embodied environments can be abstracted and represented as graphs on which to learn generalizable and interpretable policies. TextWorld environments seem a good testbed for that.

- Reducing the search space by breaking down the potential actions in groups sharing similar semantics is a neat idea. I believe this could be learned or even extracted from large language models like GTP-3.


## Concerns

- Looking at the literature around text-based games, I found relevant references that seem to be missing: Action-Elimination [1], T-DQN [2]. Since the proposed approach is trying to reduce the action space, I believe those would be decent baselines to compare against.

- It is not clear to me if the proposed approach is performing well because of the amount of prior knowledge (or expert domain knowledge as said in the paper) that has been injected into the model. Regarding the action selector, instead of using a rule-based approach, maybe some attention mechanism over the edges and nodes of the graph could be used?

- In Table 2, shouldn't there also be supporting edges that contain the "knife", e.g. the knife should be held by the player?

- In Table 3, why is the Vanilla RL approach getting 100 on training level 4, which is much higher compared to the two other models?

- Section. 5.3.3. Robustness. What happens if the edges related to the goal are dropped (e.g, the 'needs' edges from Figure 3)?

#### References
- [1] Zahavy, Tom et al. “Learn What Not to Learn: Action Elimination with Deep Reinforcement Learning.” NeurIPS (2018).
- [2] Hausknecht, Matthew J. et al. “Interactive Fiction Games: A Colossal Adventure.” ArXiv abs/1909.05398 (2020)

### Minor

 - p.3: "In text-based games, the agent receives a knowledge graph..." -> By their nature, text-based games' input modality is text. Instead, I would mention TextWorld provides the underlying state of the game represented as a graph, or something along those lines.

 - p.4: "2) Furthermore, this step also eliminates the search space to enable a feasible candidates set for the rule-miner." I don't fully understand what that means.

-----
### Typos
- p.3: The sentence "Therefore various ... models." is missing a verb.
- p.3: "... usually more than an order of magnitude smaller than original actions". I find it hard to parse "more than ... smaller". Just stating $K << |A|$ seems enough to me.
- p.3: "... action type $A_i$." -> Since $i$ refers to the time step, I don't think that makes sense to use it. Earlier, it was stated that $k = f_p(s_i)$, so $A_k$ might be more appropriate, or $A_{k_i}$ as used later in the text.
- p.4: "end-to-end differential" -> differentiable
- p.4: "we target at" -> "we aim at"?
- p.4: "After obtaned" -> "After obtaining"
- p.4: "in demonstration dataset" -> "in the demonstration dataset"
- p.4: "Therefore, we needs" -> need

**Summary Of The Paper:**

This paper is about improving the interpretability of agents trained on reinforcement learning problems where the input observation is given as a graph. Specifically, the authors propose to split the decision-making process in two: 1) an action pruner, and 2) an action selector. Experiments were conducted on a suite of TextWorld environments that have their state exposed as a knowledge graph. Empirically, it was shown that the proposed two-step approach is better at generalizing on unseen environments and is more robust to noisy perturbations on the input graphs.

**Summary Of The Review:**

Dealing with large action is an important research direction in RL. I find the proposed approach interesting, intuitive, and allows for better interpretability. While the results are encouraging, I'm concerned with the amount of prior knowledge that went into this, e.g. semantic grouping of the actions, and rule-based action selector. Could those priors be learned? Also, I believe this paper could be strengthened by evaluating their approach on other types of tasks (i.e., not related to cooking) from the TextWorld suite. For those reasons, I can't recommend it for acceptance.

---

### Official Review · Reviewer_WZ15 · 2021-11-04

**Correctness:** 3
**Technical Novelty And Significance:** 3
**Empirical Novelty And Significance:** Not applicable
**Recommendation:** 5
**Confidence:** 4

**Main Review:**

Strengths:
- The idea is novel. The proposed method provides a kind of post-hoc explanation where graph-based model with interpretability is learned based on the demonstrations generated by a decent good policy of any DRL (without consider its interpretability).
- The topic is more like a timely important topic.
- The paper is well-written.

Weakness:
- Graph-based model can provide interpretability based on causal relations, but the authors didn't talk much about this perspective regarding their own model. Instead, the authors demonstrate more interpretations to the predictive outcomes. This undermines the interpretability of the proposed method.
- Action group: according to the example of actions, it looks the current model is more suitable to handle the task like text games. I am concerned how it works if the proposed method applies to other domains, like robot control or atari games. Can you elaborate it with an example? If the proposed method currently only can handle the text games, it applicability on other domains is limited.
- regarding the empirical evaluation for generalization, robustness, it looks the baselines are selected improperly and the comparison is unfair. For example, the performance of GATA-GTF in Table 3 is bad and far from Vanilla RL, why select it as a baseline? Also, it is not suprising that Vanilla RL (as said by) doesn't have generalization ability. I also have the same questions for the comparison in Table 4.


Minors:
- "hiereachical" in sec.2.2 should be "hierarchical".
- "on one hand" in sec.3 -> "on the one hand"
- For the action type $A_k = \{ a_k^1, \ldots, a_k^n \}$ in sec.3, what does this "n" mean?


**Summary Of The Paper:**

This work targets to the problem of limited interpretability in RL. To solve the problem, the authors propose a two-step hybrid model, including action pruner and action selector. The proposed method is evaluated on the text games, with results showing its ability of interpretation, generalization, and robustness.

**Summary Of The Review:**

-The contribution of this work is primarily the empirical one.
-The authors aim at achieving the interpretability, generalization ability, and robustness of their methods. However, due to the concerns I raised in the weakness part, I prefer to weak reject (marginally below the acceptance threshold) the paper.

---

### Comment · Area_Chair_TbKR · 2021-11-27
**Authors, we are a bit surprised by the lack of response.**

We hope all is fine. Please reach out if not. I think we can still fit a very quick discussion if you submit a response today.

Thanks

---

### Decision · Program_Chairs · 2022-01-20

**Decision:**

Reject

**Comment:**

The authors did not respond to the concerns raised by all the reviewers. As the recommendation were on the edge, this lack of engagement seems odd, and it left the reviewers with little material to discuss and revise their recommendation. We recommend the authors carefully consider the reviews if they plan to resubmit.